# *Mycobacterium avium* Subspecies *Hominissuis*-Induced Fatal Vasculitis in Zebra Finches (*Taeniopygia guttata*), USA

**DOI:** 10.3390/ijms26157555

**Published:** 2025-08-05

**Authors:** Kelly Chenoweth, Carey Laster, Subarna Barua, Chengming Wang

**Affiliations:** 1Molecular Diagnostic Laboratory, Department of Pathobiology, College of Veterinary Medicine, Auburn University, Auburn, AL 36849, USA; kjc0063@auburn.edu (K.C.); szb0116@auburn.edu (S.B.); 2Thompson Bishop Sparks State Diagnostic Laboratory, Auburn, AL 36849, USA; carey.laster@agi.alabama.gov

**Keywords:** *Mycobacterium avium* subsp. *hominissuis*, zebra finch, vasculitis, acid-fast bacilli

## Abstract

*Mycobacterium avium* subsp. *hominissuis* (MAH) is a zoonotic pathogen with a broad host range and diverse clinical manifestations. We report here the first documented case of MAH-induced fatal vasculitis in zebra finch (*Taeniopygia guttata*). Histopathological examination revealed acid-fast bacilli within macrophages and endothelial cells, primarily affecting the heart and aorta. Mycobacterial DNA was detected in cloacal swabs from affected finches and environmental samples from their housing facility. PCR targeting the *rpoB* gene and insertion elements IS1245 and IS901, followed by sequencing, confirmed MAH infection. MAH DNA was identified in 4 of 13 finch cloacal swabs and 7 of 28 environmental samples. This study describes a novel, highly pathogenic manifestation of MAH in birds and underscores the potential for avian involvement in environmental and zoonotic transmission.

## 1. Introduction

*Mycobacterium avium* complex (MAC) comprises slow growing, environmentally persistent mycobacteria with clinical significance in both humans and animals [1,2]. Among the MAC members, *Mycobacterium avium* subsp. *hominissuis* (MAH) is a common cause of pulmonary and disseminated infections in immunocompromised humans and is endemic in livestock, notably pigs, where it causes granulomatous lymphadenitis [3]. Though sporadically reported in cattle, rabbits, and companion animals [4,5], avian MAH infections remain rare and poorly characterized. Avian mycobacteriosis is typically attributed to *M. avium* subsp. *avium*, with MAH limited to isolated cases in psittacines and zoo birds [6].

Here, we report the first confirmed case of MAH infection in zebra finch (*Taeniopygia guttata*), characterized by fatal vasculitis. Through histopathological and molecular analyses, we confirm MAH as the etiologic agent and propose mechanisms by which vascular inflammation contributes to the observed pathology. This report expands the known host range of MAH, highlights its pathogenic plasticity, and calls for renewed attention to birds as potential reservoirs in the epidemiology of MAC infections.

## 2. Detailed Case Description

Samples from Zebra finches used as research animals were submitted to the Thompson Bishop Sparks State Diagnostic Laboratory (Auburn, AL, USA) and Molecular Diagnostic Laboratory (College of Veterinary Medicine, Auburn University, Auburn, AL, USA) for the diagnosis of infectious disease.

Zebra finches from which the samples were submitted for disease diagnosis in this study displayed symptoms including lethargy, puffiness, and nodular lesions, and some of them exhibited hemoptysis and died. Necropsies were conducted on 8 of 23 deceased zebra finches. Ziehl-Neelsen staining revealed acid-fast bacilli within macrophages and endothelial cells, with lesions primarily in the heart and aorta (Figure 1).

The liver was seared and cut in half while looking for lesions. The liver surface was touched to the well of a Columbia blood agar plate with 5% sheep blood and MacConkey agar followed by streaking for isolation. The plates were incubated at 37 °C with 10% CO_2_ and at 37 °C aerobically for 18–24 h, respectively. After plate inoculation, half of the liver was placed in tetrathionate broth and incubated at 37 °C aerobically for 24 h. The broth was struck to both BGS and XLT4 agar and incubated at 37 °C aerobically for 18–24 h. Bacterial cultures of liver tissue failed to yield bacterial growth. 

The *rpob* gene sequences of representative *Mycobacterium* species and subspecies were retrieved from GenBank, including *M. abscessus* subsp. *massiliense* (GenBank accession number: CP065033), *M. africanum* (FR878060), *M. avium* subsp. *paratuberculosis* (CP033909, CP091845, CP171461), *M. avium* subsp. *avium* (CP046507, CP085977, CP089223), *M. avium hominissuis* (CP009360, CP018020, CP018393, CP029332, CP035744), *M. avium hominissuis* (CP035744), *M. bovis* (AP010918, LT708304, CP002095), *M. canettii* (FO203509), *M. caprae* (CP016401, AJ131120), *M. chelonae* (LR134345), *M. colombiense* (CP020821), *M. fortitum* (CP011269), *M. intracellular* (CP076378, CP003347), *M. leprae* (X53999), *M. microti* (CP010333), *M. pinnipedii* (NR025249), *M. smegmatis* (CP027541), *M. tuberculosis* (MG995115, CP046308, CP054013, CP074075, CP130774), and *M. ulcerans* (CP092429). These sequences were aligned using Clustal Multiple Alignment to identify conserved and variable regions for primer and probe design.

The following primers and probes were developed based on conserved regions: forward primer: 5′- CGCCCAACCGAGTTTCCTT -3′; reverse primer: 5′- GACAGCTCGTCGAGCACCTCTT-3′; and fluorescein probe:/56-FAM/CA GAT CGA C/ZEN/T CCT TCG AGT GGC TGA T/3IABkFQ/.

To differentiate *M. avium* subsp. *avium* from *M. avium* subsp. *hominissuis*, previously published primers were used to amplify insertion sequences IS1245 [7] and IS901 [8]. IS1245 primers: 5′-CTTGATCGACGCGGAGTTGA-3′ and 5′-AGGTGGCGTCGAGGAAGAC-3′; and IS901 primers: 5′-GCAACGGTTGTTGCTTGAAA-3′ and 5′-TGATACGGCCGGAATCGCGT-3′.

The PCR was performed on a LightCycler 480^®^II (Indianapolis, IN, USA) real-time PCR platform with a final volume of 20 µL, containing 10 µL of extracted DNA. Thermal cycling consisted of a 2 min denaturation step at 95 °C followed by 18 high-stringency step-down thermal cycles and 40 low-stringency fluorescence acquisition cycles consisting of 1 sec at 95 °C, 12 sec at 58 °C, 30 sec at 67 °C, and 10 sec at 72 °C. Melting curve analysis was performed by monitoring fluorescence between 38 °C and 80 °C.

To validate the assays, the *M. avium avium* Chester strain (ATCC) and gBlock gene fragments containing the amplicon regions of nine representative *Mycobacterium* species/subspecies were used to validate the mycobacterial PCRs used in this study. The primers, probes, and gBlock gene fragments were synthesized by Integrated DNA Technologies (Coralville, IA, USA). Based on the molarity of the gene fragment, the dilutions of the fragment were made to give solutions containing 10,000, 1000, 100, 10, and 1 gene copies/µL in T_10_E_0.1_ buffer. The products of mycobacterial PCRs were sent to ELIM Biopharmaceuticals (Hayward, CA, USA) for DNA sequencing. The nucleotide sequences were compared with those of representative *Mycobacterium* species/subspecies for identification.

Lyophilized *Mycobacterium avium* subspecies *avium* Chester, ordered from ATCC, was dissolved in one ml 1 × PBS, followed by 1:10 dilutions and 1:100 dilutions. The diluted *M. avium avium* received treatment with lysozyme (VWR; a final concentration of 18 mg/mL) or 1 × PBS, at 37 °C for one hour. Genomic DNA was performed using glass fiber matrix binding and elution with a commercial kit (High-Pure PCR Template Preparation Kit; Roche Diagnostic, Indianapolis, IN, USA) following the manufacturer’s instructions. The validated DNA extraction protocol was then applied to extract DNA from tissues and organs of the Zebra finch in this study.

The *rpob*-based PCR assay successfully amplified all representative *Mycobacterium* species with a detection limit of 10 copies per reaction. The IS1245- and IS901-targeting PCRs showed a slightly lower sensitivity (100 copies per reaction), attributed to their longer amplicon sizes.

Treatment of *M. avium avium* with lysozyme (18 mg/mL) at 37° C for one hour significantly enhanced DNA extraction efficiency. In 1:10 dilutions, copy numbers increased from 9520 to 39,833 (4.2-fold, *p* < 10^−4^), and in 1:100 dilutions, from 1156 to 5336 (4.6-fold) (Figure 1). This improvement reflects the robust cell wall of mycobacteria, which resists standard lysis methods. Lysozyme disrupts the peptidoglycan layer, allowing proteinase K and detergents to more effectively release genomic DNA.

Zebra finches from which the samples were submitted for the disease diagnosis in this study displayed symptoms including lethargy, puffiness, and nodular lesions, and some of them exhibited hemoptysis and died. Necropsies were conducted on 8 of 23 deceased zebra finches. Ziehl-Neelsen staining revealed acid-fast bacilli within macrophages and endothelial cells, with lesions primarily in the heart and aorta (Figure 2). Bacterial cultures of liver tissue failed to yield bacterial growth.

Mycobacterial PCR identified mycobacterial DNA in 4 of 13 cloacal swabs and 7 of 28 environmental or surface samples from the finch facility. PCR results were positive for IS1245 and negative for IS901, confirming the presence of MAH in the Zebra finch, but not *M. avium avium*. Sequencing of *rpob* amplicon showed 100% identity with MAH strain MAH11 (GenBank accession no. CP035744) (Figure 3).

## 3. Discussion

This study documents a novel, highly pathogenic presentation of MAH in a passerine species, manifesting as acute, fatal vasculitis—a stark contrast to the chronic granulomatous inflammation typically associated with avian MAC infections [9]. These findings suggest either an emergent, more virulent MAH strain or heightened susceptibility in zebra finches.

The observed vasculitis, rarely reported in avian MAC infections, raises the possibility of unique pathogenic mechanisms, such as endothelial invasion, strain-specific virulence factors, or an aberrant host immune response. This challenges the prevailing view that MAH predominantly affects immunocompromised mammals and reveals its adaptability to new hosts.

Detection of MAH in 25% of environmental samples is consistent with its persistence in soil and water [10]. As zebra finches are common in urban and peri-urban environments, they may act as sentinel species or bridging hosts for MAC between wildlife, livestock, and humans. Given their proximity to human dwellings, the zoonotic potential of avian MAH infections warrants further investigation, especially in the context of One Health approaches.

In this study we did not attempt to culture *Mycobacterium*, and cultures of the heart and other organs except liver were not performed. As a result, endocarditis and sepsis caused by other infectious agents could not be definitively ruled out.

This case expands the recognized host spectrum of MAH and illustrates its capacity to cause acute, lethal disease in birds. The deviation from classical granulomatous lesions suggests either evolving pathogen virulence or unique avian vulnerabilities. Broader surveillance and interdisciplinary collaboration are critical to understanding the ecology of MAH and mitigating risks posed by environmentally resilient mycobacteria.

## Figures and Tables

**Figure 1 ijms-26-07555-f001:**
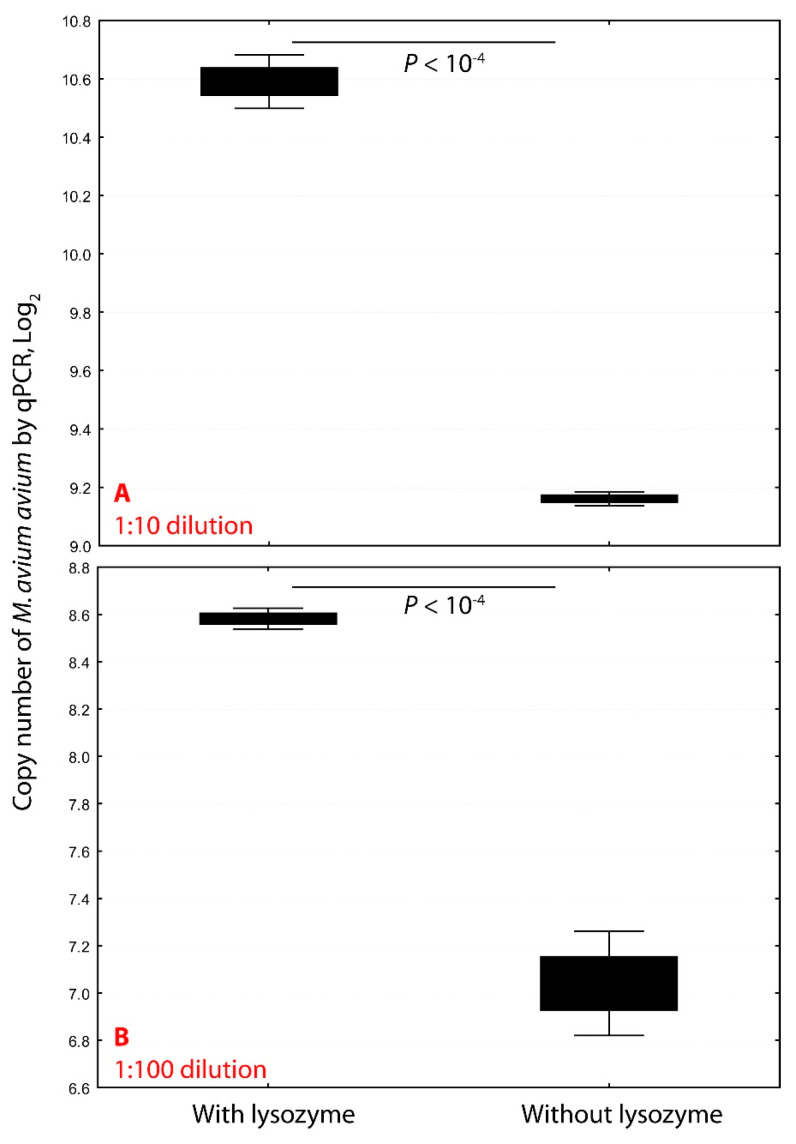
Lysozyme pretreatment significantly enhances mycobacterial DNA extraction efficiency. Pretreating *Mycobacterium avium* subsp. *avium* with lysozyme (18 mg/mL) at 37 °C for one hour markedly increased DNA yield. At a 1:10 dilution, lysozyme treatment increased the copy number 4.2-fold, from 2^9.16^ (9520) to 2^10.59^ (39,833) (*p* < 10^−4^; panel (**A**)). At a 1:100 dilution, copy number rose 4.6-fold, from 2^7.04^ (1156) to 2^8.58^ (5336) (*p* < 10^−4^; panel (**B**)). These results demonstrate that enzymatic disruption of the mycobacterial cell wall significantly improves genomic DNA recovery for downstream molecular applications.

**Figure 2 ijms-26-07555-f002:**
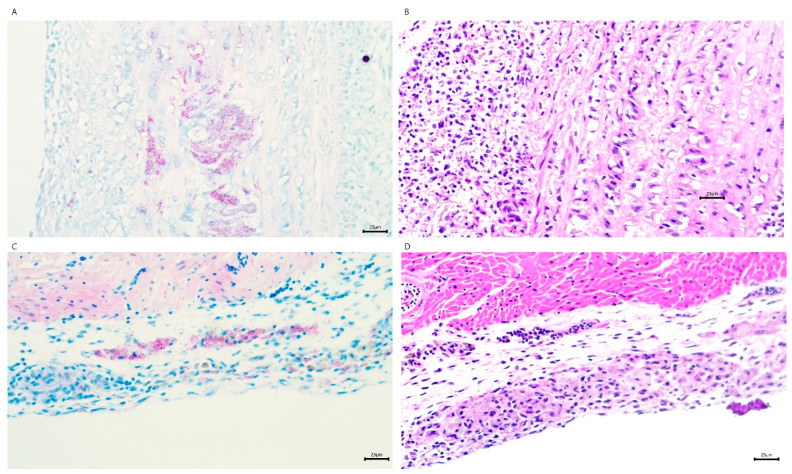
Histological characterization of *M. avium hominissuis*-induced cardiac lesions in zebra finches. (**A**) Ziehl–Neelsen (acid-fast) staining of the ascending aorta reveals granulomatous inflammation with intracellular acid-fast bacilli. (**B**) Hematoxylin and eosin (H&E) staining of a serial section adjacent to (**A**) shows granulomatous inflammation with central necrosis and a dense lymphohistiocytic infiltrate. (**C**) Acid-fast staining of the epicardium highlights granulomatous inflammation with clusters of acid-fast bacilli. (**D**) H&E staining of a serial section adjacent to (**C**) demonstrates granulomatous inflammation and associated fibrosis within the epicardium. All images were captured at 200× magnification. Acid-fast staining (**A**,**C**) confirms the presence of *M. avium* bacilli (red rods) within granulomatous lesions. Serial sections (**A**–**D**) illustrate the correlative histopathological features of mycobacterial infection.

**Figure 3 ijms-26-07555-f003:**
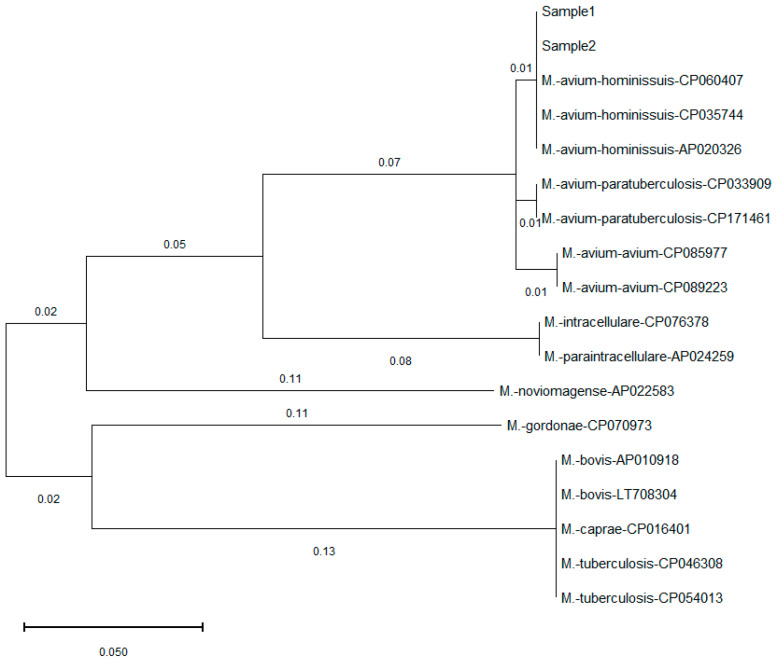
Phylogenetic comparison of *ropb* gene found in this study and published *Mycobacterium* sequences in GenBank**.** The tree is drawn to scale with branch lengths computed using the Maximum Likelihood method and measured in the number of substitutions per site. The identified *Mycobacterium* in this study (Sample 1, Sample 2) showed 100% similarity to that of *M. avium hominissuis*, and different levels of mismatches with other *Mycobacterium* species and subspecies.

## Data Availability

All data generated in this study are available from the corresponding authors upon reasonable request.

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
