# Peer review of "Mycobacterium avium Subspecies Hominissuis-Induced Fatal Vasculitis in Zebra Finches (Taeniopygia guttata), USA"

_ijms, 2025, doi:10.3390/ijms26157555_

Round 1
Reviewer 1 Report
Comments and Suggestions for Authors
In my view, before the work of Chenoweth et al. is considered suitable for publication, it needs to be radically re-written. While the introduction, discussion and abstract are reasonably well-written and easy to understand, most of the “Detailed case description section” is incoherent. This makes it close to impossible to determine if any of the conclusions are supported by the presented data.
The most promising part of the article is line 107-108: “Sequencing of the 16S rRNA amplicon showed 100% identity with MAH 107 strain MAH11 (GenBank accession no. CP035744)”. This strongly suggests that the authors have found the Mycobaterium they claim to have found, but, even so, they still need to valid this finding with phylogenetics.
The text spanning lines 76 to 92 does not make any sense at all. The text in lines 39 to 65 seems to be describing a novel 16S realtime PCR. Lines 92 to 97 seems to be evaluating a DNA extraction method. The authors need to decide if they are validating new methods or making a case report. And make sure everything they write is consistent and supporting their conclusions.
Overall, I strongly recommend that the authors find a senior researcher with experience in their field of research to help them organize their data so that it is suitable for publication in a scientific journal.
Author Response
Dear Reviewer,
Please find my response to your valuable comments in the attached PDF file. Thanks!

Reviewer 2 Report
Comments and Suggestions for Authors
This study identified a pathogenic bacterium that is innovative, but I believe there are still several areas that need improvement:
For the drug sensitivity test of pathogenic bacteria, it is necessary to supplement;
2. The experimental results of the biochemical properties of pathogenic bacteria need to be supplemented;
3. There is no ruler on the sliced image, please add it.
Author Response

(The authors gave the same response as above.)

Reviewer 3 Report
Comments and Suggestions for Authors
The submitted case report, Mycobacterium avium subspecies hominissuis -induced fatal vasculitis in Zebra finches (Taeniopygia guttata), USA, examines an interesting case of fatal tuberculosis in passerines. The vasculitis lesions are atypical of the classic presentation of avian tuberculosis and the perpetrating subspecies is unique. The case report is concise and well written. I have recommendations for improvement to the structure and Figure 1 prior to publication.
Line 84: This sentence is misplaced and should be combined with other relevant clinical details in the beginning of this section (see below regarding restructuring). Is there any additional information on the history of these zebra finches? Pets or zoo? Please include.
The detailed case description section should be restructured, as starting with extensive PCR methods is distracting. I suggest starting with lines 84-87 and 98-103, then describing PCR methodology lines 39-97, and finishing with the PCR results lines 98-108. There should be appropriate transitions between these three new sections.
Line 102: please specify how liver samples were cultured (e.g. liver homogenate was struck directly onto trypticase soy agar with 5% sheep blood and incubated aerobically for 18-24 hours).
Figure 1: “With lysozyme” legend listed twice, the right side should be without lysozyme. Also, numerical values in the figure legend do not coincide with the log scale copy number on the y axis.
General comment: Are there any gross lesion photos? I'm dying to know what these hearts looked like grossly. Also, it appears the heart was not cultured. If this is true, please make a statement that endocarditis and sepsis due to other infectious agents could not necessarily be ruled out in the discussion.
Author Response

(The authors gave the same response as above.)

Round 2
Reviewer 2 Report
Comments and Suggestions for Authors
The MS can be accepted at the present revision.